# Mitochondrial Oxidative Stress Mediates Bradyarrhythmia in Leigh Syndrome Mitochondrial Disease Mice

**DOI:** 10.3390/antiox12051001

**Published:** 2023-04-26

**Authors:** Biyi Chen, Nastaran Daneshgar, Hsiang-Chun Lee, Long-Sheng Song, Dao-Fu Dai

**Affiliations:** 1Division of Cardiovascular Medicine, Department of Internal Medicine, University of Iowa Carver College of Medicine, Iowa City, IA 52242, USA; 2Department of Pathology, Carver College of Medicine, University of Iowa, Iowa City, IA 52242, USA; 3Division of Cardiology, Department of Internal Medicine, Kaohsiung Medical University, Kaohsiung 807, Taiwan; 4Lipid Science and Aging Research Center, Kaohsiung Medical University, Kaohsiung 807, Taiwan

**Keywords:** mitochondria, oxidative stress, bradycardia, arrhythmia, Leigh Syndrome, cardiomyopathy

## Abstract

Mitochondrial oxidative stress has been implicated in aging and several cardiovascular diseases, including heart failure and cardiomyopathy, ventricular tachycardia, and atrial fibrillation. The role of mitochondrial oxidative stress in bradyarrhythmia is less clear. Mice with a germline deletion of Ndufs4 subunit respiratory complex I develop severe mitochondrial encephalomyopathy resembling Leigh Syndrome (LS). Several types of cardiac bradyarrhythmia are present in LS mice, including a frequent sinus node dysfunction and episodic atrioventricular (AV) block. Treatment with the mitochondrial antioxidant Mitotempo or mitochondrial protective peptide SS31 significantly ameliorated the bradyarrhythmia and extended the lifespan of LS mice. Using an ex vivo Langendorff perfused heart with live confocal imaging of mitochondrial and total cellular reactive oxygen species (ROS), we showed increased ROS in the LS heart, which was potentiated by ischemia-reperfusion. A simultaneous ECG recording showed a sinus node dysfunction and AV block concurrent with the severity of the oxidative stress. Treatment with Mitotempo abolished ROS and restored the sinus rhythm. Our study reveals robust evidence of the direct mechanistic roles of mitochondrial and total ROS in bradyarrhythmia in the setting of LS mitochondrial cardiomyopathy. Our study also supports the potential clinical application of mitochondrial-targeted antioxidants or SS31 for the treatment of LS patients.

## 1. Introduction

Oxidative stress has been extensively studied in various heart disease models [1]. It is widely accepted that oxidative stress due to high levels of reactive oxygen species (ROS) can cause an oxidative modification of lipids and many critical cardiac proteins, including ion channels, leading to altered ion homeostasis, Ca^2+^ overload, impaired mitochondrial function, metabolic derangement, and cell death [2,3]. Mitochondria, as the main source of ROS and the target of ROS-induced damage, have been shown to play critical roles in cardiac aging [4], cardiac hypertrophy, and heart failure with reduced or preserved ejection fractions [5,6,7]. ROS have been implicated in various ventricular tachyarrhythmias and atrial fibrillation [8,9]; however, the role of ROS in the pathogenesis of bradyarrhythmia is underexplored.

The current study aims to elucidate the mechanistic role of oxidative stress in bradyarrhythmia in mitochondrial cardiomyopathy using mice with a germline deletion of a mitochondrial complex I subunit, Ndufs4, resembling Leigh Syndrome (LS) [10]. LS is a phenotypically and genetically heterogeneous mitochondrial disease commonly presented with psychomotor regression and early mortality [11]. The most common causes are mutations in components of mitochondrial respiratory complex I [11], NADH dehydrogenase. The cardiomyopathic phenotypes of these LS mice with complex I deficiency include bradycardia due to a sinus node dysfunction and conduction defect. In this study, we showed that treatment with various mitochondrial-targeted antioxidants attenuated ROS and restored normal sinus rhythm in parallel with an improved mitochondrial membrane potential, less oxidative damage, and extension of lifespan.

## 2. Materials and Methods

For animal experiment and electrocardiography (ECG), all animal experiments were approved by the Institutional Animal Care and Use Committee (IACUC) at the University of Iowa. Germline Ndufs4^−/−^ mice (Leigh Syndrome mice) were obtained from the University of Washington. All mice were on the C57/BL6/J background and were fed the regular diet from Harlan Teklad. Both male and female mice were included (balanced gender) in this study. To treat LS mice, we inserted osmotic minipumps (Alzet 1004) loaded with SS31 (3 mg/kg/d, New England Peptide) or Mitotempo (7 mg/kg/d, Sigma Aldrich, USA) subcutaneously on the flank area under brief 1% isoflurane anesthesia. Treatment was initiated around 28–30-days-old until the end of life. ECG recordings were obtained from conscious mice for 40–60 min using the INDUS Rodent Surgical Monitoring system without anesthesia at the age of 45–48-days-old. The arrhythmia events over the whole recording period were quantified and presented as the frequency of events (per 10 min).

For Langendorff perfusion and live Imaging of ROS using confocal microscopy and simultaneous ECG recording, we modified our previously published protocols for in situ confocal imaging of T-tubules in intact hearts [12,13] to examine ROS. Briefly, the excised hearts were perfused with Kreb–Henseleit’s (KH) solution (in mM: 120 NaCl, 24 NaHCO_3_, 11.1 glucose, 5.4 KCl, 1.8 CaCl_2_, 1 MgCl_2_, 0.42 NaH_2_PO_4_, 10 taurine, 5 creatine, oxygenated with 95% O_2_, and 5% CO_2_) containing Mitosox (2 uM) and CM-H2DCFDA (DCF, 10 µM, Molecular Probes, USA) for 30 min through a retrograde Langendorff perfusion system at room temperature. After fluorescent probe loading was completed, the hearts were then transferred to another Langendorff apparatus for perfusion at 37 °C. We positioned the heart inside a recording chamber attached to LSM510 confocal microscope (Carl Zeiss MicroImaging Inc., Oberkochen, Germany) equipped with 63x (NA = 1.4) oil immersion lens. In situ fluorescence images of Mitosox and DCF were acquired from left ventricular epicardial myocytes under sinus rhythm. Blebbistatin (10 μM, Sigma) and BDM (2, 3-butandion-monoxim, 10 mM, Sigma) were added to the perfusion solution to prevent motion artifacts during imaging. TMRM or DCF fluorescence was tandemly excited at 488 and 561 nm wavelength, and the emission wavelength was 500–545 and > 575 nm, respectively. After 10 min stabilization, the baseline images were acquired; then, the perfusion solution was switched to a mock ischemia solution (in mM: 135 NaCl, 5.4 KCl, 1.8 CaCl, 1 MgCl, 0.33 NaH_2_PO_4_, 10 Hepes) without 95% O_2_ and 5% CO_2_ for 30 min. The 512 × 512 frame images were acquired at baseline, then 1, 3, 5, 10, 15, 20, 30, 45, and 60 min after reperfusion/reoxygenation with KH solution. The optical pinhole was set to 1 airy disc (<1 µm axial resolution) during confocal imaging. Every frame image contained 202 × 202 µm^2^ area of left ventricular epicardial myocytes. Simultaneous Pseudo ECG recording was performed using epicardial leads. Five images from different locations of each left ventricular free wall were acquired, and intensity values from each ventricle were averaged to represent the global Mitosox and DCF fluorescence intensity of each ventricle using Image J, then plotted over time after reperfusion.

Three experimental groups were used for Langendorff perfusion and ROS studies (n = 3 each group), including WT, Ndufs4^−/−^, and Ndufs4^−/−^ mice pre-treated with Mitotempo (7 mg/kg) ~6 h prior to the experiments.

For immunohistochemistry for oxidative damage, to evaluate oxidative damage, we performed immunohistochemistry using anti nitro-tyrosine primary antibody (Millipore #06-284), followed by anti-mouse IgG-HRP (Abcam, MA, US; ab97046) secondary antibody and DAB staining. Quantitative analysis was performed using Fiji Image J by color deconvolution method.

For cell culture, for the in vitro studies, human embryonic kidney 293 (HEK293) cells (ATCC, Manassas, VA, USA) cultured in DMEM medium supplemented with 10% FBS and 1% Pen Strep were used. Ndufs4 gene was deleted using the CRISPR/Cas9 method.

For generation of Ndufs4 knock-out cells using the CRISPR/Cas9 method, HEK293 cells with 70–80% confluency were used for transfection according to well established protocols. In brief, cells were transfected with 2.5 µg of guideRNAs targeting Ndufs4 (NF4SgRNA, Addgene) mixed with 5 μL of Lipofectamine 2000 (Invitrogen). After 48 h of transfection, the cells were exposed to Geneticin™ Selective Antibiotic (G418 Sulfate) at 40 µg/mL, and selection was performed for approximately 7 days. Subsequently, cells were serially diluted and cultured at a density of 5 cells per well of 96-well plate. In the next couple of weeks, colonies were picked and expanded for further confirmation of successful deletion using Western blotting.

For live cell staining and immunostaining, cells were plated on glass-bottom dishes. Culture medium containing DCFDADA (5 mM) and TMRE (25 nM) were added to dishes and incubated for approximately 30 min, followed by 5 min incubation with Hoechst 33342 in new medium without the fluorescent probes. We took multiple confocal images using a Leica SP8 confocal microscope using optimized spectra as guided by Molecular Probes Spectraviewer.

For statistics, all statistical analyses and calculations were performed using Stata IC version 10 and plotted using GraphPad Prism (version 8.0; GraphPad Software, Inc., CA, USA). The results are reported as the mean ± standard error of the mean. To test for significant difference between groups, we applied either Student’s *t*-test, nonparametric Kruskal–Wallis test, or analysis of variance (ANOVA) with Sidak post hoc test for comparison whenever appropriate. Kaplan–Meier method was used for survival curves analysis, and log-rank test was applied to compare survivor functions between groups. A *p*-value less than 0.05 was considered statistically significant.

## 3. Results

### Leigh Syndrome Mice Develop Bradyarrhythmia

The Ndufs4^−/−^ Leigh Syndrome (LS) mice had normal left ventricular mass (Figure 1A) and Ejection Fraction (Figure 1B) as measured by echocardiography in conscious mice without an anesthetic agent. LS mice developed profound bradycardia with a heart rate of 384.5 ± 17.8 bpm, significantly lower than that in WT mice (580 ± 35, *p* < 0.0002, Figure 1C). The heart weight normalized to tibia length were similar between WT and LS mice. Both LV mass and heart weight data do not show any evidence of hypertrophic cardiomyopathy that has been reported in ~10% of LS patients [14].

We performed electrocardiograms (ECGs) from conscious immobilized Ndufs4^−/−^ mice to understand the characteristics of bradycardia. As shown in Figure 2, the spectrum of bradyarrhythmia in Ndufs4^−/−^ mice includes frequent sinus node dysfunction (SND, Figure 2A), such as sinus pauses and sinus bradycardia (Figure 2B); episodic atrioventricular (AV) conduction block that manifested as Mobitz type II second-degree AV block (Figure 2C); or rarely, a high-grade AV block (Figure 2D). A quantitative ECG analysis of these mice demonstrates frequent bradyarrhythmia with a median [interquartile range] frequency of 62 [39, 76], in contrast to the complete absence of arrhythmia in the WT littermates (Figure 2H). Treatment with either mitochondrial antioxidant Mitotempo (7 mg/kg/d) or mitochondrial protective peptide SS31 (3 mg/kg/d), delivered by subcutaneous Alzet 1004 minipumps initiated around 28–30-days-old, significantly decreased arrhythmia events in Ndufs4^−/−^ mice (frequency of 2.7 [1, 16] and 4.4 [4.2, 27.6], respectively, Figure 2F–H).

To elucidate the mechanistic role of ROS in causing bradycardia in the setting of complex I mitochondrial cardiomyopathy, we performed a confocal scanning of ex vivo Langendorff-perfused hearts (Figure 3) loaded with Mitosox (an indicator of mitochondrial superoxide) and 2’,7’-dichlorodihydrofluorescein diacetate (DCFDA, a marker of total cellular H_2_O_2_) with a simultaneous ECG recording. The Ndufs4^−/−^ hearts had mild non-significant increases in mitochondrial superoxide (higher Mitosox) and total cellular H_2_O_2_ (higher DCFDA) at baseline (Figure 3E,M,N). The signals of both Mitosox and DCFDA fluorescent indicators in Ndufs4^−/−^ hearts were dramatically potentiated by simulated hypoxia reoxygenation injury (H/R, Figure 3F). These were much higher than the signals in WT hearts after reoxygenation (Figure 3B,M,N). An intraperitoneal injection of the mice with Mitotempo 6 h before the experiment abolished the increase in both mitochondrial and total ROS, both at baseline (Figure 3I) and after reoxygenation (Figure 3J,M,N). Simultaneous epicardial ECG recordings (Figure 3Q) revealed the sinus rhythm in WT hearts (Figure 3C,D) and an episodic AV block and sinus arrhythmia (arrow) in Ndufs4^−/−^ hearts at baseline (Figure 3G). This bradycardia progressed into a complete AV block after H/R (Figure 3H). Mitotempo pre-treatment in these mice corrected arrhythmia both at baseline and after H/R, restoring it to normal rhythm (Figure 3K,L). These findings suggest that an acute increase in mitochondrial and total cellular ROS may represent one mechanism for a conduction block in the context of mitochondrial complex I deficiency. A fluorescence intensity measurement demonstrates that, compared with WT hearts, Ndufs4^−/−^ hearts have significantly higher ROS (higher fluorescence intensity) at baseline or soon after reoxygenation (Figure 3M,N). Interestingly, while the ROS increase is transient during H/R in WT hearts, we observe a persistent increase in ROS in Ndufs4^−/−^ hearts after H/R injury (Figure 3M,N). The measurement of the area under the curve (AUC) of fluorescence signals over post-reoxygenation time show a greater than 2-fold increase in both Mitosox and DCFDA in Ndufs4^−/−^ hearts, which were prevented by Mitotempo (*p* < 0.0001, Figure 3O,P).

As increased oxidative stress is expected to cause oxidative damage, we performed immunostaining of cardiac left ventricular sections using anti-nitrotyrosine antibodies. The Ndufs4^−/−^ left ventricles displayed increased nitrotyrosine compared with WT left ventricles, indicating increased protein oxidative damage (Figure 4A,B,D). Chronic treatment with Mitotempo significantly attenuated 3-nitrotyrosine staining in the cardiomyocytes (Figure 4C,D). The survival analysis shows a significant extension of the median (20%) and maximal (16%) lifespans in Ndufs4^−/−^ mice treated with continuous SS31 treatment in subcutaneous minipumps (Figure 4E red curve, log-rank *p* = 0.03). Likewise, there is a significant extension of the median (29%) and maximal (18%) lifespans in Ndufs4^−/−^ mice treated with Mitotempo, delivered in subcutaneous minipumps (Figure 4E green curve, log-rank *p* < 0.01). These findings suggest that a chronic increase in mitochondrial and total cellular ROS represents an important mechanism of cardiac arrhythmia, the molecular pathogenesis of complex I deficiency, and the survival of Ndufs4^−/−^ mice.

HEK293 cells have been widely used in electrophysiological (EP) research [15] as a heterologous system to study EP properties of the exogenous cardiac ion channel. To confirm our findings in an ex vivo heart (Figure 3) in a separate cell type, we used the CRISPR/Cas9 method to delete Ndufs4 in HEK 293 cells, as confirmed by a Western blot (Figure 5A). Live cell staining showed increased DCFDA, indicating higher total cellular ROS levels in Ndufs4KO cells (Figure 5C,F). Conversely, tetramethylrhodamine ethyl ester (TMRE), a marker of mitochondrial membrane potential, were significantly lower in Ndufs4 KO cells (Figure 5G), suggesting the drop in mitochondrial membrane potentials. These findings suggest that increased ROS and impaired mitochondrial function in the context of Ndufs4 deficiency is conserved in multiple cell types. The drop in TMRE and increased DCFDA was ameliorated by either Mitotempo or SS31 (Figure 5D–G).

## 4. Discussion

In this study, we show evidence of the direct mechanistic role of mitochondrial oxidative stress in bradycardia using mitochondrial disease LS mice, which were generated from a germline homozygous deletion of exon 2 of the encoding gene Ndufs4 (Ndufs4^−/−^). While encephalopathy is the main manifestation in LS patients, various manifestations of mitochondrial cardiomyopathy were identified in 18–21% of these LS patients, and cardiac involvement is associated with a worse prognosis. Hypertrophic cardiomyopathy is the most common abnormality in these patients [14,16]. In a multicenter study of 130 Leigh Syndrome patients, Sofou et al. [14] reported that cardiac dysfunction was present in 23 patients (17.7%), of which 12 had hypertrophic cardiomyopathy (9.2%). Arrhythmia or conduction defects were reported in 5 patients and dilated cardiomyopathy in 2 patients.

Mitochondrial diseases are well known to have protean clinical manifestations (i.e., certain mitochondrial syndromes predispose the patient to distinct abnormalities [17]). Because of the high ATP demands of the heart, patients with a mitochondrial disease are susceptible to cardiac involvement. For examples, Kearns–Sayre syndrome (KSS) is characterized by chronic progressive external ophthalmoplegia, abnormal pigmentary retinopathy, and cardiomyopathy and/or progressive arrhythmia leading to a complete heart block. The atrioventricular conduction defects of KSS may present as syncope or sudden death, and these occur after the symptoms of retinopathy and ophthalmoplegia. Myoclonic epilepsy with ragged red fibers (MERRF) and mitochondrial encephalopathy with lactic acidosis and stroke-like episodes (MELAS) may also involve the heart with cardiac hypertrophy, dilated cardiomyopathy, and arrhythmia. In a recent large study of cardiac outcomes in 600 adults with genetically confirmed mitochondrial diseases, Savvatis et al. reported that cardiac involvement was present in 38.3%, including LVH (22.5%), heart failure with reduced ejection fraction (~7%), and a conduction defect (9.3%). The latter also includes a shortened PR interval (5.2%) and Wolff–Parkinson–White syndrome (3.4%). During the first 10 years after the initial assessment, ~11.7% developed a major adverse cardiovascular event, including ~6.2% having the major cardiac arrhythmia events [17].

Ndufs4^−/−^ (LS) mice have features of growth retardation, lethargy, cerebellar ataxia, loss of motor function due to both encephalopathy and myopathy, hypothermia, slowed breathing, and apnea [10,18]. The encephalomyopathy features of LS mice are explicit and unambiguous; however, cardiac involvement has been inconsistent, similar to variable (~20%) cardiac involvement in LS patients. The myocyte-specific loss of Ndufs4 (driven by CKM-NLS-Cre) displayed features of hypertrophic cardiomyopathy [19]. In contrast, the cardiomyocyte-specific loss of Ndufs4 (driven by αMHC) did not show significant structural abnormalities or systolic dysfunction but aggravated pressure-overload-induced heart failure [20]. This was associated with cardiac protein hyperacetylation, alterations in the redox state, and inhibition of Sirt3 activity [20]. In the absence of cardiac hypertrophy, the current germline Ndufs4^−/−^ (LS) mice showed severe sinus node dysfunction and intermittent AV conduction block (reported in ~5% in LS patients). These discrepancies can be explained by a well-known heterogeneity in various mitochondrial diseases. For example, LS has been linked to mutations in >50 different genes encoding mitochondrial respiratory complexes I, III, IV, and V, etc. Conversely, the mutation in one gene (e.g., Ndufs4) can manifest with different clinical syndromes in patients: encephalopathy, myopathy, and/or cardiomyopathy, with or without lactic acidosis, indicating a wide range of the clinical spectrum.

Here, we applied the state-of-the-art method: the ex vivo Langendorff perfused heart, coupled with live and continuous ROS imaging of beating hearts using a confocal microscope, and a simultaneous ECG recording. This method excludes CNS regulation of the heart rate or the potential confounding effects of metabolic acidosis on heart rhythm. Increased ROS has been well accepted as a main mechanism of a reperfusion injury [21]. Consistently, our simulated hypoxia-reoxygenation significantly increased mitochondrial superoxide (Mitosox) and total cellular H_2_O_2_ (DCFDA fluorescence) in WT hearts, which was observed within minutes of reoxygenation and peaked at 20 min. This mild and transient increase in ROS did not affect the heart rhythm, as simultaneous ECG showing normal sinus rhythm. In contrast, confocal scanning of epicardial cardiomyocytes in ex vivo LS Ndufs4^−/−^ hearts showed higher (but not-significantly higher) mitochondrial and total ROS at baseline. These higher ROS signals were associated with mild sinus node dysfunction and intermittent AV block. In Ndufs4^−/−^ hearts, simulated H/R injury substantially increased both mitochondrial and total ROS. An ECG showed a concomitant progression to a high degree AV block. Pretreatment with Mitotempo, a direct scavenger of mitochondrial superoxide, significantly abolished Mitosox and DCFDA signals, suggesting that mitochondria are the main sources of ROS in the context of complex I deficiency. The fact that Mitotempo completely restored the simultaneous ECG to normal sinus rhythm provides robust evidence of the direct mechanistic role of mitochondrial ROS in causing sinus node dysfunction and AV block in the context of mitochondrial cardiomyopathy. Our recent study reported that the conduction tissue-specific loss of Ndufs4 (driven by HCN4-Cre) in mice is sufficient to cause similar arrhythmia in a cell-autonomous manner, although the metabolic defect of Ndufs4^−/−^ in neurons, cardiomyocytes, and the other cell types in the hearts (e.g., inflammatory cells [22,23]) could also contribute to arrhythmia by impacting the neurohormonal or paracrine regulation of cardiac rhythm [24].

The Na_v1.5_ cardiac Na^+^ channel (encoded by the *SCN5A*) is one of the most important ion channels for cardiomyocytes and SA node excitability and impulse propagation through the AV node, which can be modified by ROS. There are two plausible downstream mechanisms of our findings. First, mitochondrial ROS induced by exogenous NADH has been shown to reduce *I*_Na_ (Na_v1.5_ current) in a HEK-293 heterologous system [15]. We demonstrated the use of HEK-293 cells that Ndufs4 deletion impaired mitochondrial membrane potential/respiration leading to increased ROS, which may reduce Na_v1.5_ current. Second, reducing mitochondrial ROS, either directly by Mitotempo scavenging of mitochondrial superoxide or indirectly through cardiolipin-related mechanisms (by SS31) may attenuate the decreased NAD^+^/NADH in the context of the Ndufs4 deletion [24]. This partial restoration of NAD^+^/NADH by mitochondrial antioxidant can be achieved through nicotinamide nucleotide transhydrogenase (NNT) to supply NAD^+^ and through inhibition of glutathione reductase (GR) that consumes NAD^+^ (Figure 6). Our recent study showed that one mechanism of decreased *I*_Na_ is via hyperacetylation at K1479 Na_v1.5_ due to the impairment of NAD^+^-dependent Sirtuin 1 deacetylase [24].

Interestingly, Zhang et al. reported that MHC-Ndufs4 knockout ameliorates ischemia-reperfusion injury in Langendorff perfused heart. Lower mitochondrial respiration measured by TMRE in Ndufs4-deficient cardiomyocytes may prevent the burst of ROS during reperfusion (measured by mt-HyPer and MitoSox). This was explained by a decrease in ROS generated by complex I-dependent reverse electron transport [25]. Different from our current study using germline Ndufs4^−/−^ (LS mice), the MHC-Ndufs4^−/−^ had a normal cardiac function at baseline, and the Ndufs4 deletion is limited to cardiomyocytes, which is not as severe as in LS mice. The contribution of mitochondrial dysfunction from non-myocytes (e.g., macrophages, fibroblasts) may aggravate the phenotypes seen in LS [22,23]. Furthermore, different levels of mitochondrial damage might contribute to the difference in the mitochondrial electron transfer chain and ROS production between germline and cardiomyocyte-specific knock-out models. We propose that the disruption or imbalance of supply and demand in terms of ATP, NAD^+^/NADH, and Glutathione antioxidant system within the mitochondria might explain the well-known heterogeneity and protean manifestations mitochondrial disease. In other words, whether abnormal phenotypes are observed or not depends on cell-types, microenvironment, and context (rest vs. under stress).

Increased ROS in LS hearts were confirmed by increased 3-nitrotyrosine. The 3-nitrotyrosine is a biomarker of oxidative damage that is formed by nitration of protein-bound and free tyrosine residues by reactive peroxynitrite molecules. We further showed that continuous treatment with either Mitotempo or SS31 significantly extended survival. SS31 (also known as Elamipretide) highly concentrates within mitochondria by binding to cardiolipin (Figure 6), which is highly enriched in the mitochondrial inner membrane [26]. It preserves cristae ultrastructure and enhances the formation of super-complexes, thereby facilitating the electron transfer through mitochondrial respiratory complexes [27]; it also prevents cardiolipin from converting cytochrome c into a peroxidase, all of which decrease the leakage and indirectly decreased ROS [28]. Biochemical studies revealed that SS31 alters the distribution of ions at the interface, modulates bilayer physical properties, and reduces the energetic burden of calcium stress in mitochondria [29]. A chemical cross-linking with mass spectrometry study identified 12 proteins interacting with SS31, including some respiratory complex subunits in the electron transfer and ATP production, as well as 4 proteins involved in the 2-oxoglutarate process [30]. It has an excellent safety profile as it has minimal effect on cardiac proteome in normal wild-type mouse hearts [5].

Several factors may contribute to the mortality of LS mice, including apneic breathing patterns [18], severe bradycardia [24], and progressive neurological symptoms [10]. One limitation of our current study is the use of a small molecule approach that has a systemic effect and is not able to answer the cell-type specificity of our treatment, but it may facilitate the translation to clinical therapeutics. Treatment options for LS patients are very limited; however, promising pre-clinical studies using the same germline Ndufs4^−/−^mouse model have identified potential new therapies to alleviate LS pathologies, including rapamycin [31], chronic hypoxia [32], supplementation of NAD^+^ [24], and microglial ablation [33]. Chronic hypoxia decreased toxic oxygen metabolites and has been shown to improve survival, body weight, body temperature, behavior, neuropathology, disease biomarkers, and brain NAD^+^ concentration in LS mice [34]. In these studies, chronic hypoxia was achieved by housing LS mice in a hypoxic chamber with 11% O_2_ [32] or inducing severe anemia with a therapeutic phlebotomy or inhaling a controllable low-dose of carbon monoxide [35]. Although the hypoxic approaches sound interesting, the translation of these therapeutics to clinical use may not be feasible [35]. We and others have recently reported that NAD^+^ supplementation [24,36] or microglial ablation by Pexidartinib [33,37], an FDA-approved drug, had beneficial effects in alleviating both encephalopathy and cardiomyopathy in LS mice.

## 5. Conclusions

In summary, the current study provides robust evidence of the direct mechanistic roles of ROS in bradyarrhythmia in LS. It also suggests that SS31, a tetrapeptide currently tested in multiple phases 2/3 clinical trials for various mitochondrial diseases, has a great translational potential for the treatment of LS.

## Figures and Tables

**Figure 1 antioxidants-12-01001-f001:**
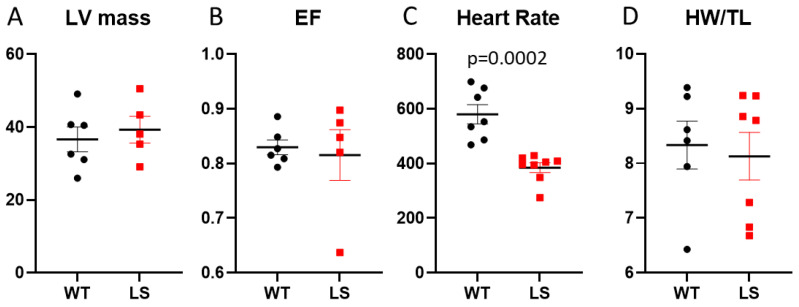
Echocardiographic measurement of (**A**) LV mass, (**B**) ejection fraction (%), (**C**) heart rate (bpm) in WT and Leigh Syndrome (LS, Ndufs4^−/−^) mice, and (**D**) heart weight normalized to tibia length (mg/mm); n = 5–9. No significant difference except for heart rate, *p* = 0.0002 by *t*-test.

**Figure 2 antioxidants-12-01001-f002:**
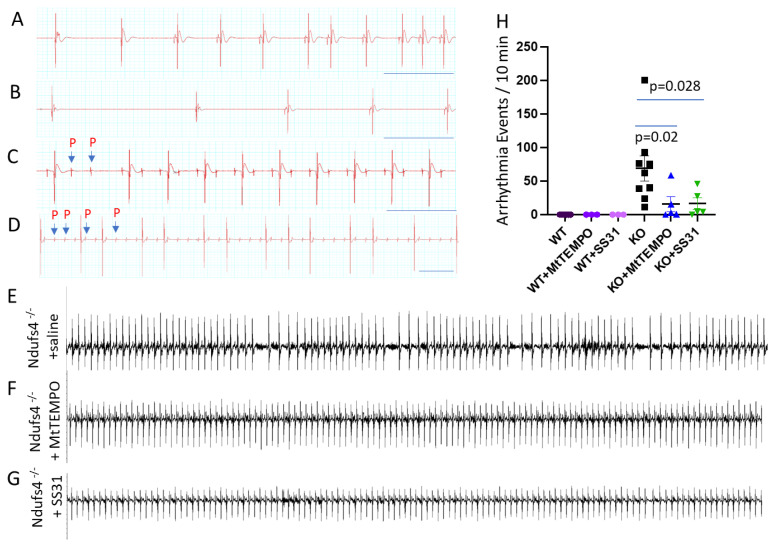
(**A**–**D**) The spectrum of bradyarrhythmia in Ndufs4^−/−^ mice includes sinus node dysfunction (**A**), long sinus pauses (**B**), second degree AV block (**C**), and high-grade AV block (**D**). Arrows: non-conducted P waves; scale bar: 0.5 sec. (**E**–**G**). Representative ECG tracings show frequent bradyarrhythmic events in € saline treated Ndufs4^−/−^ mice, relatively normal ECG in Ndufs4 ^−/−^ mice treated with (**F**) Mitotempo or (**G**) SS31 administered by subcutaneous osmotic pumps. (**H**) Quantitative analysis of in vivo arrhythmia events in Ndufs4^−/−^ or WT mice treated with saline, Mitotempo, or SS31 between 45–50-days-of-age; n = 5–9. Scale bar: 0.5 s.

**Figure 3 antioxidants-12-01001-f003:**
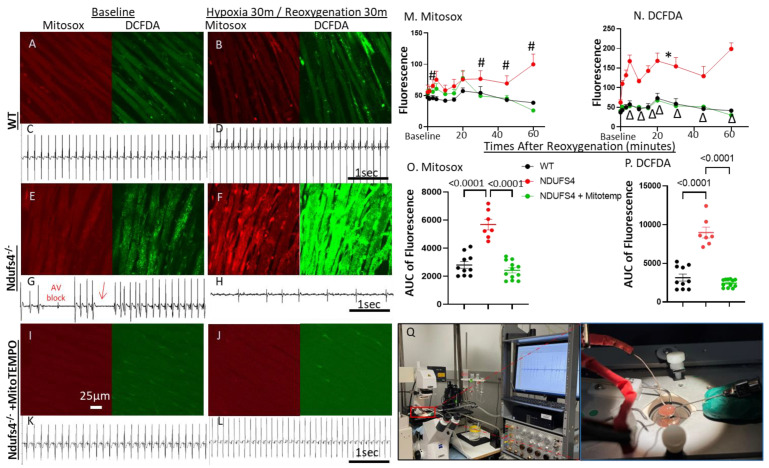
Confocal scanning of ex vivo Langendorff-perfused hearts loaded with Mitosox and DCFDA show increased ROS in Ndufs4^−/−^ hearts (epicardial cardiomyocytes) (**E**) at baseline (**E**,**M**–**P**), which was substantially augmented after simulated hypoxia reoxygenation, H/R injury (**F**,**M**–**P**), when compared to WT heart (**A**,**B**). ROS is abolished by Mitotempo treatment (**I**,**J**). Simultaneous epicardial ECGs show sinus rhythm in WT heart (**C**,**D**), episodic AV block and sinus arrhythmia (arrow) in Ndufs4^−/−^ hearts at baseline (**G**), and high-grade AV block (**H**) after reoxygenation, concomitant with dramatic increases in both Mitosox and DCF (**F**). (**I**–**L**) Mitotempo treatment abolished Mitosox and DCF and restored normal sinus rhythm. (**M**) Mitosox, (**N**) DCFDA fluorescence intensity after reoxygenation, (**O**) area under the curve of Mitosox fluorescence intensity, (**P**) area under the curve of DCFDA fluorescence intensity; n = 3; #, *p* < 0.05 Ndufs4 vs. both WT and Mitotempo; * *p* < 0.05 vs. WT; ∆ *p* < 0.05 Ndufs4 vs. Mitotempo for each time point. (**Q**) Photo of our ex vivo Langendorff perfused heart system, equipped with confocal microscopy for live imaging and simultaneous epicardial ECG recording. Right panel image shows the platform of microscope and epicardial leads.

**Figure 4 antioxidants-12-01001-f004:**
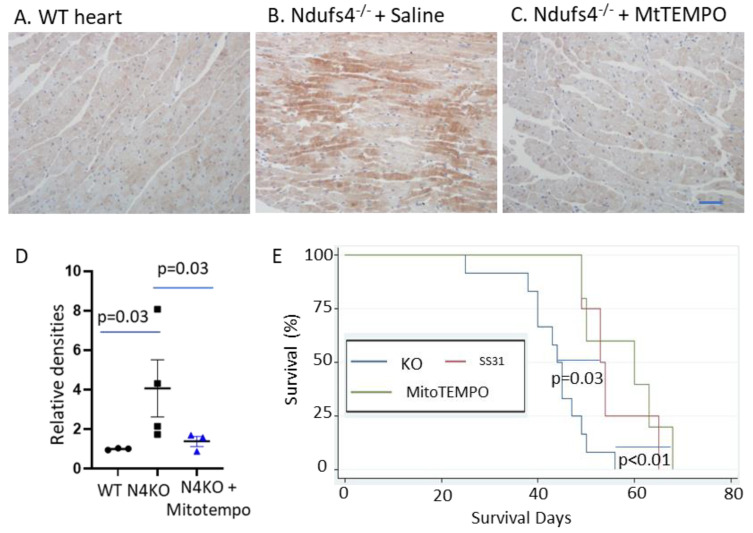
Representative immunohistochemistry (IHC) for 3-nitrotyrosine in (**A**) WT, (**B**) LS Ndufs4^−/−^ heart, (**C**) Mitotempo-treated Ndufs4^−/−^ hearts, (**D**) Analysis of relative densities of IHC; n = 3–4 for mouse left ventricles IHC by non-parametric tests. (**E**) Survival analysis shows significant lifespan extension in Ndufs4^−/−^ mice treated with either SS31 (red) or Mitotempo (green) compared with saline-treated Ndufs4KO mice (blue); n = 4–11, by log-rank tests. Scale bar: 50 µm.

**Figure 5 antioxidants-12-01001-f005:**
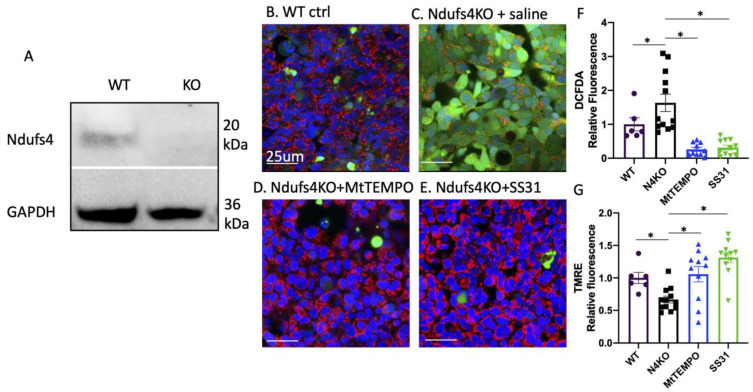
(**A**) Western blot of WT-HEK293 cells and Ndufs4 KO- cells. Live confocal imaging of Ndufs4-deleted HEK293 (**C**–**E**) and control cells (**B**) loaded with TMRE and DCFDA shows increased ROS (green, DCFDA) and decreased TMRE (loss of red) in Ndufs4-deleted cells (**C**). Treatment with (**D**) Mitotempo or (**E**) SS31 restored TMRE and ROS (**F**,**G**); n = 6–12 for cells; * *p* < 0.05 by ANOVA and post hoc analysis.

**Figure 6 antioxidants-12-01001-f006:**
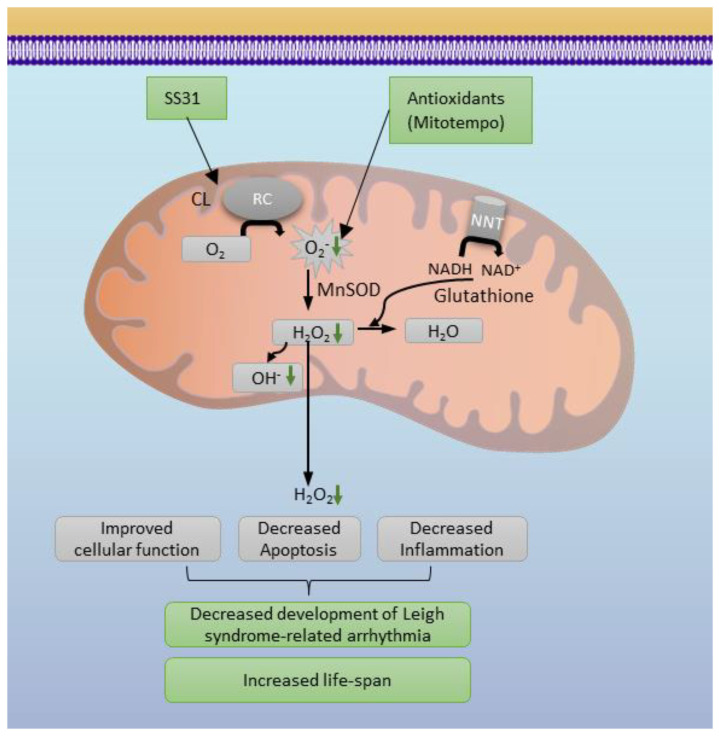
Mechanistic diagram of mitochondrial and cytosolic ROS in LS cardiomyopathy. RC: respiratory complexes, CL: cardiolipin, NNT: Nicotinamide Nucleotide Transhydrogenase.

## Data Availability

The data presented in this study are available in each figure, as individual datapoint. No big data is generated in the current study.

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
