# Peer review of "Mitochondrial Oxidative Stress Mediates Bradyarrhythmia in Leigh Syndrome Mitochondrial Disease Mice"

_antioxidants, 2023, doi:10.3390/antiox12051001_

Round 1

Reviewer 1 Report

The paper by Chen et al. describes increased oxidative stress in Ndufs4 deficient LS mice. The paper is interesting and investigations on cardiac phenotypes in mitochondrial disease are worth publishing, as these are severe life threatening conditions with significant clinical heterogeneity in human patients. The data presented is convincing and supports the conclusions, however I have some questions and comments that need to be addressed before  publication.

First of all, there are significant details missing from the methods part. Most importantly, did the authors study male or female mice or were both genders present in the study groups? This should be clearly stated somewhere in the text and the different groups should be comparable in relation to the biological sex of the analyzed mice.

Secondly some of the methods lack important  details, like the CRISPR/Cas treatment of the HEK cells. How was the Cas9 delivered to the cells, the guideRNA plasmid data is incomplete, how was the selection performed, which drug at what concentration etc.  Also, no data is presented on validation of the results from this treatment, so it is impossible to comment on the success of the treatment.

Overall the conclusions are supported by the data, however in two separate occasions, rows 281-282 and 333-335, the authors speculate that heteroplasmy level may affect the outcome of the clinical disease in mice and in patients. This is not true for nuclear gene mutations like Ndufs4, and only influences subunits encoded by mitochondrial DNA.

I would also wish that the authors would discuss a bit whether Bradyarrhythmia is seen in human patients with LS or other primary mitochondrial disease and the relevance of this to human patients.

And some explanation on the mechanism of action of the used antioxidants would be beneficial. The image on figure 5 is not very informative, it would at least need some explanation in text form.

Minor comments:

Abstract row 13, aging is listed as cardiovascular disease, this is a bit strange.

Row 15 Ndufs4subunit 

Row 86-87 same sentence repeated twice

Figures 3 and 4, scale bars are missing from the images

Row 382  onwards data availability and acknowledgements seem to be the generic text from the manuscript template.

Author Response

The paper by Chen et al. describes increased oxidative stress in Ndufs4 deficient LS mice. The paper is interesting and investigations on cardiac phenotypes in mitochondrial disease are worth publishing, as these are severe life-threatening conditions with significant clinical heterogeneity in human patients. The data presented is convincing and supports the conclusions, however I have some questions and comments that need to be addressed before publication.

First of all, there are significant details missing from the methods part. Most importantly, did the authors study male or female mice or were both genders present in the study groups? This should be clearly stated somewhere in the text and the different groups should be comparable in relation to the biological sex of the analyzed mice.

Reply: Both male and female mice (balanced gender) were studied. This has been added to the method section.

Secondly some of the methods lack important details, like the CRISPR/Cas treatment of the HEK cells. How was the Cas9 delivered to the cells, the guideRNA plasmid data is incomplete, how was the selection performed, which drug at what concentration etc. Also, no data is presented on validation of the results from this treatment, so it is impossible to comment on the success of the treatment.

Reply: As suggested, we have added more details of the method, and add a validation figure to demonstrate the successful ablation of Ndufs4 in cells by using Western blot (Figure 5A).

Generation of Ndufs4 knock-out cells using the CRISPR/Cas9 method. HEK293 cells with 70-80% confluency were used for transfection, according to well-established protocols. In brief, cells were transfected with 2.5 μg of guideRNAs targeting Ndufs4 (NF4SgRNA, Addgene) mixed with 5 μl of Lipofectamine 2000 (Invitrogen). After 48 hours of transfection, the cells were exposed to Geneticin™ Selective Antibiotic (G418 Sulfate) at 40 μg/mL, and selection was performed for approximately 7 days. Subsequently, cells were serially diluted and cultured at a density of 5 cells per well of 96-well plate. In the next couple of weeks, colonies were picked and expanded for further confirmation of successful deletion using western blotting.

Overall the conclusions are supported by the data, however in two separate occasions, rows 281-282 and 333-335, the authors speculate that heteroplasmy level may affect the outcome of the clinical disease in mice and in patients. This is not true for nuclear gene mutations like Ndufs4, and only influences subunits encoded by mitochondrial DNA.

Reply: We agree with the reviewer that heteroplasmy is known for mitochondrial DNA mutation. We have removed it in those two places, to avoid any confusion.

I would also wish that the authors would discuss a bit whether Bradyarrhythmia is seen in human patients with LS or other primary mitochondrial disease and the relevance of this to human patients.

Reply: We have added one paragraph to discuss this:

“Mitochondrial diseases are well known to have protean clinical manifestations, i.e. certain mitochondrial syndromes predispose to distinct abnormalities(17). Because of the high ATP demands of the heart, patients with mitochondrial disease are susceptible to cardiac involvement. For examples, Kearns-Sayre syndrome (KSS) are characterized by chronic progressive external ophthalmoplegia, abnormal pigmentary retinopathy, and cardiomyopathy and/or progressive arrhythmia leading to complete heart block. The atrioventricular conduction defects of KSS may present as syncope or sudden death, and these occur after the symptoms of retinopathy and ophthalmoplegia. Myoclonic epilepsy with ragged red fibers (MERRF) and mitochondrial encephalopathy with lactic acidosis and stroke-like episodes (MELAS) may also involve the heart with cardiac hypertrophy, dilated cardiomyopathy and arrhythmia. In a recent and large study of cardiac outcomes in 600 adults with genetically confirmed mitochondrial diseases, Savvatis et al reported that cardiac involvement was present in 38.3%, including LVH (22.5%), heart failure with reduced ejection fraction (~7%) and conduction defect (9.3%). The latter also includes a shortened PR interval (5.2%) and Wolff-Parkinson-White syndrome (3.4%). During the first 10 years from initial assessment, ~11.7% developed a major adverse cardiovascular event, including ~6.2% having the major cardiac arrhythmia events(17).”

And some explanation on the mechanism of action of the used antioxidants would be beneficial. The image on figure 5 is not very informative, it would at least need some explanation in text form.

Reply: As suggested, we have expanded the discussion on the potential mechanisms of SS31 since it is a promising drug currently tested for multiple mitochondrial diseases. “SS31 (also known as Elamipretide) highly concentrates within mitochondria through binding to cardiolipin, which is highly enriched in the mitochondrial inner membrane(26). It preserves cristae ultrastructure and enhances super-complexes formation, thereby facilitating the electron transfer through mitochondrial respiratory complexes(27); it also prevents cardiolipin from converting cytochrome c into a peroxidase, all of which decrease the leakage and indirectly decreased ROS(28). Biochemical studies revealed that SS31 alters the distribution of ions at the interface, modulates bilayer physical properties, and reduces the energetic burden of calcium stress in mitochondria(29). A chemical cross-linking with mass spectrometry study identified 12 proteins interacting with SS31, including some respiratory complex sub-units in the electron transfer and ATP production, as well as 4 proteins involved in the 2-oxoglutarate process(30).”

Minor comments:

Abstract row 13, aging is listed as cardiovascular disease, this is a bit strange.

Reply: We have revised “Mitochondrial oxidative stress has been implicated in aging and several cardiovascular diseases, including heart failure and cardiomyopathy, ventricular tachycardia, and atrial fibrillation.”

Row 15 Ndufs4subunit

Row 86-87 same sentence repeated twice

Figures 3 and 4, scale bars are missing from the images

Row 382 onwards data availability and acknowledgements seem to be the generic text from the manuscript template.

Reply: These have been revised as suggested.

Reviewer 2 Report

In this manuscript Authors use an animal model of Leigh Syndrome to study the role of mitochondrial oxidative stress in cardiac bradyarrhythmia and in other correlated cardiac problems . Two mitochondrial-targeted antioxidants were used to analyze their efficacy in fighting cardiomyopathy.

In my opinion, this work is well conceived, was carried out by using a number of independent approaches and is correctly discussed

Author Response

We greatly appreciate the positive feedback from this reviewer.

Reviewer 3 Report

Chen et al have done an interesting study on the role of mitochondrial oxidative stress on cardiac bradyarrythmia. They have been using a transgenic mice with germline deletion of Ndufs4subunit respiratory complex I which develops severe mitochondrial encephalomyopathy resembling Leigh syndrome (LS). These mice display several types of cardiac bradyarrhythmia including sinus node dysfunction and episodic atrioventricular (AV) block. The authors were able to demonstrate that treatment with the mitochondrial antioxidant Mitotempo or mitochondrial protective peptide SS31 significantly ameliorated bradyarrhythmia and extended the lifespan of LS mice. Using ex-vivo Langendorff perfused heart with live confocal imaging of mitochondrial and total cellular reactive oxygen species (ROS), they were also able to show that the ischemia-induced  increased ROS in LS heart was abolished with Mitotempo  treatment.  Oxidative damage in cardiac tissue was also reduced in cardiac tissue from Mitotempo-treated Ndufs4-/- mice.   Using the a HEK293 cell line, the authorus were also able to show that mitochondrial membrane potential was reduced Ndsfs4KO cells together with increased ROS, this was abrogated with Mitotempo or SS31 treatment of the cells.

The manuscript in generally well written, the methods used are impressing and the findings are interesting.

Major issues

I find the figures and the lay-out very busy and they could be amended to be easier to follow. Also, some of the statements regarding differences between groups do not refer to quantified data.

Why is oxidative damage in the hearts (Figure 4 G-I) together with cell work? Would not this be more natural to follow the chronic animal study in general (Figure 1) although you only have three groups (why not SS31 in this analysis?). I recon these data are from the chronic animal study – or are they from the Langendorf study? Please explain. I also find Figure 5 A to be more natural to follow the chronic animal data. Could Figure 5 B stand alone in the discussion section together with the text? Please consider.

The authors have not included a group of wild-type mice treated with Mitotempo or SS31. Do you have any preliminary data showing no effects in WT mice, please explain why you chose to leave out these groups both in animals and in cells.

 Specific comments

Line 164-1168: The Ndufs4-/- hearts had increased mitochondrial superoxide (higher Mitosox) and increased total cellular H2O2 (higher DCFDA) at baseline (Figure 3E). The signals of both Mitosox and DCFDA fluorescent indicators in Ndufs4-/- hearts were dramatically potentiated by simulated hypoxia reoxygenation injury (H/R, Figure 3F).

Figure 3E is a representative picture of baseline Mitosox and DCFA flouresence. In order to refer to an increase in ROS, this needs to be quantified and tested. Where is the quantification of ROS and arrythmia events at baseline? The same holds for 3F – you need to refer to quantified data from baseline values.

 Line 169-170. Intraperitoneal injection of the mice with Mito-TEMPO 6 hours before the experiment abolished the increase in both mitochondrial and total ROS, both at baseline (Figure 3I) and after reoxygenation (Figure 3J). This protocol is not described in the method section and should be included. Does this hold for all Langendorf studies, or are they done with miniosmotic pumps? Please explain.

Line 198-191: Live cell staining with tetra methylrhodamine ethyl ester (TMRE), a marker of mitochondrial membrane potential and DCFDA showed significantly lower TMRE in Ndufs4 KO cells, indicating that mitochondrial membrane potential was decreased (Figure 4B).

Again, please do not refer to a representative picture when talking about significant changes, should be Figure 4 E and F.

Please go through the text and make sure that the findings you refer to are referring to quantified data in the result section.

Please refrain from referring to figures in the discussion text.

Line 350-351, please rephrase “using the same mouse model” to using the specific model used (sound like it is a patient model).

Author Response

Major issues

I find the figures and the lay-out very busy and they could be amended to be easier to follow. Also, some of the statements regarding differences between groups do not refer to quantified data.

Why is oxidative damage in the hearts (Figure 4 G-I) together with cell work? Would not this be more natural to follow the chronic animal study in general (Figure 1) although you only have three groups (why not SS31 in this analysis?). I recon these data are from the chronic animal study – or are they from the Langendorf study? Please explain.

Reply: We appreciate this suggestion and we have rearranged figures: the oxidative damage and survival data (now figure 4) to follow the animal study.

 I also find Figure 5 A to be more natural to follow the chronic animal data. Could Figure 5 B stand alone in the discussion section together with the text? Please consider.

Reply: We agree with this suggestion and have rearranged the figures to improve the flow.

The authors have not included a group of wild-type mice treated with Mitotempo or SS31. Do you have any preliminary data showing no effects in WT mice, please explain why you chose to leave out these groups both in animals and in cells.

Reply: As suggested by the reviewer, we have added these control groups and as expected, we do not observe any arrhythmia events in the WT groups treated with Mitotempo or SS31. In our previous study, SS31 is shown to be safe without any significant effect in WT mouse heart proteome (ref 5).

 Specific comments

Line 164-1168: The Ndufs4-/- hearts had increased mitochondrial superoxide (higher Mitosox) and increased total cellular H2O2 (higher DCFDA) at baseline (Figure 3E). The signals of both Mitosox and DCFDA fluorescent indicators in Ndufs4-/- hearts were dramatically potentiated by simulated hypoxia-reoxygenation injury (H/R, Figure 3F). Figure 3E is a representative picture of baseline Mitosox and DCFA flouresence. In order to refer to an increase in ROS, this needs to be quantified and tested. Where is the quantification of ROS and arrhythmia events at baseline? The same holds for 3F – you need to refer to quantified data from baseline values.

Reply: The ROS quantification at baseline was presented as time 0 in the first draft. Since this is confusing and not very appropriate, we have revised the “0” to baseline. Indeed, these were not significant, and the manuscript has been revised accordingly.

 Line 169-170. Intraperitoneal injection of the mice with Mito-TEMPO 6 hours before the experiment abolished the increase in both mitochondrial and total ROS, both at baseline (Figure 3I) and after reoxygenation (Figure 3J). This protocol is not described in the method section and should be included. Does this hold for all Langendorff studies, or are they done with miniosmotic pumps? Please explain.

Reply: This has been clarified in the method section, under the Langendorff perfusion

“Three experimental groups were used for Langendorff perfusion and ROS studies (n=3 each group), including WT, Ndufs4-/-, and Ndufs4-/- mice pre-treated with Mitotempo (7 mg/kg) ~6 hours prior to the experiments.”

This was not done with osmotic pumps. It was an acute treatment.

Line 198-191: Live cell staining with tetra methylrhodamine ethyl ester (TMRE), a marker of mitochondrial membrane potential and DCFDA showed significantly lower TMRE in Ndufs4 KO cells, indicating that mitochondrial membrane potential was decreased (Figure 4B).

Again, please do not refer to a representative picture when talking about significant changes, should be Figure 4 E and F. Please go through the text and make sure that the findings you refer to are referring to quantified data in the result section.
Reply: We appreciate this comment. This has been revised accordingly.

Please refrain from referring to figures in the discussion text.

Reply: Agree.

Line 350-351, please rephrase “using the same mouse model” to using the specific model used (sound like it is a patient model).

Reply: This has been specified.

Reviewer 4 Report

In this manuscript, the authors show that in the Leigh syndrome model mouse, generated by knocking out Ndufs4, oxidative stress is associated with bradyarrhythmia, and removal of ROS reduces arrhythmia in these mice. The authors conclude that ROS produced due to Ndufs4 deficiency plays a role in bradyarrhythmia.

The manuscript is well-written and well-presented.

Author Response

(The authors gave the same response as above.)

Round 2

Reviewer 3 Report

Major comments

The manuscript is far better in the revised version, however, looking through it, there are still some revisions that needs to amended, see minor comments for examples.

The authors should be careful in stating changes when they are not significant. This aspect is improved in this version of the manuscript, but I still think it should be more improved. Ex: looking at Figure 3 M and N, to me there is no difference between WT and the KO hearts, however the differences in ROS become obvious when these hearts are subjected to cardiac stress such as hypoxia-reperfusion (Figure 3). The immunohistochemistry done on cardiac tissue do however suggest that these hearts have been exposed increased oxidative stress over time in vivo. Line321-325, please clarify – is there difference in ROS at baseline or not? The findings in this study is clear, focus on the significant changes.

Minor comments.

Please check all abbreviations and that you have them uniformly throughout the manuscript. In some cases, they are lacking, only use abbreviations that are further used in the text.

The text have different font in figure 3 – ex: <0.0001

Figure 4, please add what the IHC is– 3-nitrotyrosine? What cardiac tissue? Left ventricular sections? Atrial?- is of in figure text and figure legend – please correct Figure.4.

Line 299-303. Please rephrase – two sentences are in one.

Line 315..please rephrase. Effect of ..on heart rhythm

Line 352-356, please rephrase and break into two sentences.

Line 366-368. Comment:  Yes, your data from the confocal experiments could definitely suggest that cardiac stressors could lead to difference in the development of cardiac dysfunction.
